# Low-Value Clinical Practices: Knowledge and Beliefs of Spanish Surgeons and Anesthetists

**DOI:** 10.3390/ijerph17103556

**Published:** 2020-05-19

**Authors:** Jesús María Aranaz Andrés, José Lorenzo Valencia-Martín, Jorge Vicente-Guijarro, Cristina Díaz-Agero Pérez, Nieves López-Fresneña, Irene Carrillo, José Joaquín Mira Solves

**Affiliations:** 1CIBER Epidemiología y Salud Pública (CIBERESP), Servicio de Medicina Preventiva y Salud Pública, Hospital Universitario Ramón y Cajal, IRYCIS, 28034 Madrid, Spain; jesusmaria.aranaz@salud.madrid.org; 2Servicio de Medicina Preventiva y Salud Pública, Hospital Universitario Ramón y Cajal, IRYCIS, 28034 Madrid, Spain; jose.valencia@salud.madrid.org (J.L.V.-M.); cristina.diazagero@salud.madrid.org (C.D.-A.P.); nieves.lopez@salud.madrid.org (N.L.-F.); 3Servicio de Medicina Preventiva y Salud Pública, Hospital Universitario Ramón y Cajal, Departamento de Medicina y Especialidades Médicas, Facultad de Medicina, Universidad de Alcalá, IRYCIS, 28034 Madrid, Spain; 4Miguel Hernández University of Elche, 03202 Elche, Spain; icarrillo@umh.es (I.C.); jose.mira@umh.es (J.J.M.S.); 5Foundation for the Promotion of Health and Biomedical Research in the Valencian Region (FISABIO), 46020 Valencia, Spain; 6Alicante-Sant Joan Health District, Ministry of Health, 03550 Alicante, Spain; 7REDISSEC, Health Services Network Oriented to Chronic Diseases, Spain

**Keywords:** medical overuse, unnecessary procedures, surgery, anesthesia

## Abstract

*OBJECTIVES*: To know the frequency and causes of low value surgical practices, according to the opinion of surgeons and anesthetists, and to determine their degree of knowledge about the Spanish “Choosing wisely” initiative. *METHODS*: Cross-sectional observational study, based on a self-administered online questionnaire through an opportunistic sample of 370 surgeons and anesthetists from three Spanish regions, contacted through Scientific Societies. The survey took part between July and December 2017. *RESULTS*: A patient profile requesting unnecessary practices was identified (female, 51−65 years old and unaffiliated disease). The frequency of requests was weekly or daily for 50.0% of the professionals, of whom 15.1% acknowledged succumbing to these pressures. To dissuade the patient, clinical reasons (47%) were considered the most effective. To increase control and safety in the case was the main reason to indicate them. The greatest responsibility for overuse was attributed to physicians, defensive medicine and mass media. Assessing professionals’ knowledge on unnecessary practices, an average of 5 correct answers out of 7 was obtained. Some 64.1% of the respondents were unaware of the Spanish “Choosing wisely” initiative. *CONCLUSION*S: Low value surgical practices are perceived as a frequent problem, which requires an approach entailing intervention with patients and the media as well as professionals. Increase awareness on unnecessary surgical practices, and how to avoid them remain essential.

## 1. Introduction

Overuse is defined as the provision of a service whose potential harm exceeds a possible benefit [1]. In the study of this phenomenon, several terms have been used, such as overtesting [2], overscreening [3], overdiagnosis [4], overprescribing [5], and overtreatment [6]. It is a field of study that has become increasingly important in recent years, which is reflected in a significant increase in scientific publications that address this issue [7], also called low-value clinical practices.

All health provision has a cost and an inherent risk of an incident related to patient safety. However, in overuse, this risk is also unnecessary. According to the literature, up to 40% of the medical practices studied may not offer clinical benefits [8], with oncology being one of the specialties in which this phenomenon has been most studied [4]. Low-value clinical practices also have a negative impact on the efficiency of the health system, as it involves a loss of opportunity cost. In the United States, during 2011, the cost related to unjustified treatments was between 158–226 billion dollars [9]. In Spain, for example, 2–3% of patients who undergo knee arthroplasty do not need such an intervention and subsequently develop complications, such as infections or cardiovascular events [10]. In addition, overuse can also have psychological repercussions on patients. For example, a false positive in an unjustified breast cancer screening could lead to emotional damage for the user, as well as numerous added diagnostic and therapeutic interventions that are totally unnecessary [11].

For decades, various institutions have developed work areas that investigate the frequency and characteristics of unjustified variations in different clinical practices, such as The Healthcare Fact Check: Regional Variations in German Health Care [12] (Germany, 1977), The Institute for Clinical Evaluative Sciences [13] (Canada, 1994), The Dartmouth Atlas of Health Care [14] (United States, 1996), and the Atlas of Variations in Medical Practice [15] (Spain, 2001). Following this line of work, numerous initiatives have emerged in recent years aimed at reducing the performance of clinical practices that do not add value. Among them are Choosing Wisely [16], promoted by The American Board of Internal Medicine (ABIM) Foundation; and Do not do [17], developed by The National Institute for Health and Care Excellence (NICE), in the United Kingdom. In Spain, the Ministry of Health, Social Services, and Equality launched the “Commitment to the Quality of Scientific Societies” (ICC) [18] initiative in 2013. In this case, more than 40 scientific societies collaborated, each of which selected the five most relevant unnecessary clinical practices of each medical specialty, following a methodology similar to that employed by the Choosing Wisely initiative [16].

Overuse affects both developed countries and those with fewer economic resources [10] and has been evaluated in different ways, through a review of medical records, by the variation of spending and health outcomes among similarly developed countries [19], or by analyzing the complaints and claims made by patients [20]. This problem has also been analyzed by directly asking professionals their opinion and assessment. In Italy, 55% of physicians considered overuse as a relevant problem [21]; in the United States, 21% of health practices have been considered unnecessary [6]; and in China, 19% of doctors said that they practice defensive medicine on a regular basis [22].

This study aimed to estimate the frequency and factors associated with overuse in the surgical field in our country, according to the perception of surgical specialists in our environment.

## 2. Materials and Methods 

This was a cross-sectional descriptive study based on a structured questionnaire answered online and aimed at professionals in surgical specialties and anesthesia practicing in a hospital in the regions of Madrid, Valencia, and Murcia that were active between July and December 2017.

This study was developed in the context of the “No Hacer” initiative (“Do-not-do”, the Spanish “Choosing Wisely” initiative) launched by the Spanish Ministry of Health, Social Services, and Equality. In addition, it is part of a project to characterize the phenomenon of overuse in the surgical field, considering its frequency, characteristics, and determinants; as well as the economic impact on potentially related claims.

To estimate the population proportions with a confidence level of 95%, a minimum required sample size of 267 participants was calculated, assuming a 5% error, accuracy of 6%, and a population frequency of 50%.

As an inclusion criterion, it was established that the participant was a practicing professional in a position of doctor of a surgical specialty or anesthesia (either specialist with a degree or in training). No exclusion criteria were established. An intentional or convenience sampling was carried out, through the Spanish Association of Surgeons, the Spanish Society of Orthopedic Surgery and Traumatology, and the Spanish Society of Anesthesiology, Resuscitation, and Pain Therapy. Potential participants received an email with an invitation signed by the project’s principal investigator, which included information about the study and the completion of the form and ensured the anonymity of the participation. The questionnaire was self-administered through the Google Forms [23] platform between July and December 2017.

The questionnaire used was based on the one designed by the research group SOBRINA for a homologous study previously carried out with the participation of primary care professionals [24,25,26]. The group of researchers in this study reviewed the questions and adapted them to the context and clinical reality of surgeons and anesthetists, verifying the degree of understanding of the questions. In the questionnaire section referring to unnecessary procedures most requested by patients, more than one response could be marked, and a free text field was added. In the section referring to the predominant patient profile requesting unnecessary procedures, the option “Do not know/No answer” and a free text field were added. On the rest of the form, only one response per section was allowed.

In order to assess the degree of knowledge about the recommendations integrated in the “Do not do” set of the ICC [18], 7 statements related to common clinical practices in the surgical field were prepared by the research team, which the respondent had to qualify as true or false and the number of rights and wrongs was measured.

### Analysis Plan

The questionnaire was designed in such a way that each respondent had to answer all the questions. The responses collected in the free text fields were recoded to other existing values or new categories were created when necessary. Internal consistency was examined by comparing the answers obtained between various related sections.

A descriptive analysis was made by calculating the frequency estimators (percentages, means, and standard deviation) with their respective 95% confidence intervals (95% CI). The analysis was performed globally (on the total of the participating sample) and stratified according to the gender and work experience of the participants.

We assessed associations between the patient profile (sex, age, and clinical situation) referred by the respondents as those requesting unnecessary procedures with the most requested therapeutic procedures, frequency of situations related to overuse, causes of overuse, and effectiveness of the arguments to convince the patient that the test or procedure he is requesting is unnecessary.

The association between ICC knowledge with the total number of correct answers to the questions that measured knowledge on recommendations integrated in the “Do not do” set was analyzed globally and individually. The answers to the questionnaire were also analyzed according to the number of questions answered correctly, considering a success rate of 70% as a cut-off point (2 or less erroneous questions out of 7).

For the comparison of qualitative variables, we used Chi square test, and Fisher’s exact test when the analysis did not fulfil the assumption for this parametric test. For the comparison of means, the Student’s t test for independent samples was used, after checking the normality assumptions with the Shapiro–Wilk test (*p* value > 0.05). For all the frequency estimates, the 95% confidence intervals (α = 0.05) were estimated, as well as the *p*-value of significance. Differences with a *p*-value less than 0.05 were considered statistically significant. The statistical exploitation of the data was carried out using statistical software Stata^®^ v.13 [27] (StataCorp LLC).

## 3. Results

A total of 370 doctors responded, the majority being men (224; 60.5%) with 16 or more years of professional experience (244; 65.9%). Only one of the participants reported having less than one year of work experience.

### 3.1. Frequency and Perceived Causes of Overuse

Table 1 summarizes the frequency of requests for unnecessary procedures by patients, as well as the answers usually given by physicians. In total, 50.0% of professionals (185) reported receiving requests for tests or procedures deemed unnecessary on a weekly or daily basis. In addition, 15.1% (56) acknowledged indicating, with the same frequency, some unnecessary procedure due to patient pressure, compared with 57.3% (212) who stated that they also convinced patients that such requests were inappropriate. In total, 26.8% (99) reported negative or aggressive reactions when trying to explain to patients the reasons why these requests were unnecessary. No differences were observed according to the professionals’ experience. The referred absence of negative reactions by the patient was significantly more frequent in those professionals with more than 15 years of work experience (38.9% versus 21.4%, *p* = 0.002).

The main reasons why the professionals indicated unnecessary clinical procedures were ‘to increase control and safety on the case’ (48.1%; 95% CI 43.0 to 53.2) and because of ‘persistent pressure by the patient’ (34.6%; CI95 29.9 to 39.6). No significant differences were observed according to the gender or the work experience of the professionals. (Figure 1). In total, 29.3% (29 of 99) who reported a negative reaction ‘weekly or daily’ after reasonably rejecting an inadequate request considered more frequently the ‘fear of being sued’ as a reason for an indication of overuse, compared to 11.4% of those who reported these reactions with a ‘monthly’ or lower frequency (*p* < 0.001).

The highest degree of responsibility in overuse (Figure 2) was attributed to the ‘physicians’ themselves (7.6 points out of 10) and defensive medicine (7.5 points), followed by the ‘press, radio, and television media’ (7.0 points). ‘On the contrary, the professionals of ‘nursing and other health professions’ were the least related to this responsibility (4.2 points). Significant differences were observed according to the gender or experience of the professionals taking part in the survey.

### 3.2. Profile of the Patient Who Demands Unnecessary Tests or Interventions

According to the perception of the professionals, those patients who most frequently requested unnecessary tests were women (59.5%; 220); and patients aged between ‘51 and 65 years’ (43.2%; 161) and ‘31 to 51 years’ (33.2%; 123). The most demanding clinical profiles were ‘patient with a disease not yet identified’ (33.0%; 122), ‘patient consulting on the Internet’ (28.1%; 104), and ‘the multi-pathological patient’ (15.1%; 56).

Figure 3 details the unnecessary procedures most requested by patients, which were mainly diagnostic techniques (67.3%; 775), followed by therapeutic procedures (18.8%; 217) and referrals to other specialists (13.9%; 160). Two of the professionals surveyed reported that “patients did not request unnecessary tests”. The most requested unnecessary procedures were computerized axial tomography (CAT), magnetic resonance imaging (MRI), routine check-ups, referral to other specialists, and the use of antibiotic therapy. According to the experience of the professionals, significant differences were observed in the request for antibiotic therapy (reported most frequently by professionals with ≤ 15 years of experience, *p* = 0.001) and in the request for bone densitometry (most frequently referred to by those with ≥ 16 years of experience, *p* <0.001). No significant differences were observed according to the gender of the professionals.

### 3.3. Most Effective Arguments for Discouraging Inappropriate Requests from Patients

The arguments considered most effective by professionals to convince the patient about the inadequacy of medical procedures were ‘clinical reasons based on knowledge’ (46.8%; 95% CI 41.7 to 51.9), ‘patient’s own safety’ (42.7%; 95% CI 37.7 to 47.8), and ‘the obtaining of the same result by other tests or previous procedures’ (39.7%; 95% CI 34.8 to 44.8) (Figure 4). No significant differences were observed according to the gender or the work experience of the professionals. This last argument, however, was considered very effective by 40.0% of the professionals who identified ‘man’ as the profile of the most frequent applicant, compared to 16.4% of the professionals who considered women as the most demanding patient (*p* = 0.017).

### 3.4. Knowledge of the ‘Commitment to Quality’ Strategy

In total, 64.1% (237) of the respondents reported not knowing the ICC campaign, this ignorance being more frequent in those with 16 years of work experience (41.0%, compared to 26.2% of the rest of professionals, *p* = 0.005). The estimated usefulness of an educational campaign to reduce the requirement of patients for unnecessary medical tests and procedures was 7.9 points out of 10, with significant differences according to the gender of the professionals taking part in the survey (8.2 in women, and 7.7 in men, *p* = 0.038).

The average of the correct answers regarding questions related to unnecessary surgical procedures selected by surgical scientific societies in the ICC was 5 out of 7, with better results in those with 15 years of work experience (average of 5.3 correct questions versus 4.8 correct answers in the rest of the professionals, *p* = 0.002), and in those who reported knowing the ICC (5.2 compared to 4.8 of those who did not know this initiative, *p* = 0.010).

The questions with lower proportions of correct answers (Table 2) were the procedure of informed consent (41.9% of correct answers; 95% CI 36.9 to 47.0) and the indication of prostate-specific antigen (50.5% of correct answers; 95% CI 45.4 to 55.6). Significant differences were found in the knowledge of peri-surgical antibiotic prophylaxis, according to the gender of the professional (80.1% in women, compared to 64.7% in men, *p* = 0.001), and according to work experience (84.9% in those with ≤ 15 years of experience, compared with 63.5% of the rest, *p* < 0.001).

## 4. Discussion

The perspective of surgical doctors on overuse has hardly been studied in our country. Among these specialists, overuse is perceived as a phenomenon with multifactorial determinants, among which the role of physicians themselves (excessive control and defensive medicine) stand out but also the pressure of patients and the media.

Approximately half of the surgeons routinely receive requests from their patients for indicated unnecessary clinical procedures, mainly diagnostic procedures (especially radiological tests), as well as the prescription of antibiotic therapies or referral to other specialists. Other authors have also described how these kinds of pressures influence clinical judgment and patient management [28]. In our study, however, most doctors dissuaded the patient with arguments based on clinical grounds and the impact on patient safety, consistent with what has been highlighted in other studies [29], and only rarely do surgeons end up succumbing to and indicating an unnecessary practice.

In relation to the type of requests made by patients, the results are consistent with previous studies, in which diagnostic tests are the most demanded [4,30], giving rise to unnecessary CT or MRI scans [31]. In the United States, 37% of patients in primary care consultations requested unnecessary diagnostic procedures, while 97% of doctors working in the emergency department indicated, at least once, an unnecessary CT scan or MRI. Diagnostic imaging tests also constitute 32% of the diagnostic procedures recently investigated in the field of health care overuse. The profile of the patient requesting unnecessary procedures corresponds to a woman aged 51–65 years, similar to that obtained by a study that characterized the overuse of radiological tests [32], and with a disease not yet identified, a profile similar to that described in primary care [24].

In Italy [21], overuse has been associated with greater security and fear of the legal consequences of their clinical decision, although in a somewhat lower percentage than the American Medical Association study, in which almost 9 out of 10 physicians surveyed were concerned regarding the possibility of litigation [6]. This study confirms the relationship between overuse and defensive medicine, although it adds pressure from patients as another determinant cause of overuse, in line with other studies [6,24]. 

The results of this study with respect to the practice of defensive medicine are at a midway point between the 93% of the physicians surveyed from surgical, emergency, and radiology specialties in Pennsylvania (United States) who declared that they carried out practices related to defensive medicine [33] and the 19% of the doctors surveyed in China [22]. In addition, defensive medicine is given a greater weight as a determinant factor of overuse, almost 10 percentage points of what was considered in a similar study by primary care physicians [24], which is probably due to the greater frequency of judicial claims against surgical professionals [34].

It is striking how often everyday surgical procedures, such as the administration of informed consent, the indication of antibiotic therapy, or the removal of body hair in the surgical field, are still clinical practices in which some surgical professionals of our environment have areas of improvement with regard to knowledge, especially among the most experienced professionals. This makes us reflect on the strategy of the prevention of unnecessary clinical practices adopted in the ICC, which probably ought to be accompanied by a dissemination and evaluation strategy among professionals in each related health care area. Less than 4 out of 10 participants reported knowing the ICC, a frequency similar to that reported by other Spanish studies, 24% and almost 15% higher than that observed in surgeons in Massachusetts (with respect to the “Choosing Wisely” initiative) [35]. In our study, the knowledge of the ICC also conditions the frequency of correct answers in relation to surgical overuse practices. Among primary care professionals, a similar result was obtained using questions adapted to their usual medical practice [25]. The fact that, in our study, those respondents with ≤15 years of professional experience more correctly answered these questions could be explained by having a more recent and updated academic training. 

An educational campaign aimed at the population, in view of these results, is valued as positive to reduce the pressure on doctors and avoid overuse, in accordance with the findings of other studies [31]. Other proposals were also considered, such as specific training of resident physicians to reduce overuse due to this cause [6], or approaching media managers to discuss their role in health information, especially information on the Internet, which is also considered partly responsible for overuse, especially when considering the volume of traffic and information on this subject on the network [36,37,38]. 

Although the recommendations established by Clinical Practice Guidelines (CPGs) are theoretically based on the best available scientific evidence, up to 40% of the sanitary procedures they recommend may not offer a net clinical benefit or even cause more harm to the patient than the clinical practices they replaced [8]. In addition, health professionals’ adherence to these recommendations also depends on factors related to their clinical experience, training, patient characteristics, and work environment [39]. Therefore, for CPGs to be a useful tool to reduce overuse, it is necessary to reflect deeply on their contents and how to implement them in clinical practice.

The design of our study presents some limitations that must be considered in the interpretation of its results. In the first place, the sample was recruited by non-randomized convenience sampling, with voluntary participation, which could condition a greater participation of more sensitized physicians, overestimating the frequency and degree of knowledge about unnecessary practices and ICC. This design, nevertheless, allowed quite a complete sample to be obtained when estimating the phenomenon in three different Spanish regions. On the other hand, our results are limited to a sample of medical professionals in the surgical field; it would be of great interest to be able to investigate all these aspects in other healthcare areas and health professionals in order to fully understand the phenomenon of overuse. In addition, it is necessary to consider possible cultural differences that influence the determinants of overuse, with respect to the results reported by other studies.

On the other hand, there is no standardized instrument to estimate the phenomenon of overuse, which is why a questionnaire has been prepared through an expert consensus inspired on that employed by the ABIM Foundation to analyze overuse in the United States [35]. This questionnaire is similar to the one developed by our group in another previous study on this phenomenon in the primary care setting, which allows us to establish comparisons between both levels of care [24]. The results obtained, however, are not directly comparable with other studies conducted in countries with different models of health and legal organization, which can significantly condition both the perception and attitudes of professionals and patients in relation to this phenomenon.

The web design of the survey facilitated the accessibility of all potential participants. The restriction on unanswered questions made it possible to avoid the presence of lost data in the subsequent analysis of the results, and the introduction of some answers in free text format enhanced the exploratory nature of some of the questions asked. The analysis of internal coherence in the sections related to defensive medicine has shown consistent results that suggest a remarkable quality in the completion of the questionnaire.

The questions designed to assess knowledge about the adequacy of clinical procedures were inspired by the main overuse procedures selected by the main surgical scientific societies in the ICC. However, they estimate knowledge, and it is not possible to know its application in the usual clinical practice of the respondents.

To our knowledge, this study is the first one to estimate the perceptions, attitudes, and knowledge about the most frequent low value surgical practices in Spain. It was carried out with a methodology comparable to previous studies carried out on this phenomenon [21,35] assessing low-value clinical practices perception and attitudes in primary care [24,25].

Our results allow us to draw useful conclusions for the orientation of resources and information awareness campaigns that allow a reduction of the overuse of health care practices in the surgical field, which should be addressed to both professionals and patients. The perceptions of physicians can serve as a first guide to know which strategies are more effective at reducing overuse, although it is necessary to carry out further studies to examine more specific dimensions of the problem from both the patient’s and the health care professional’s perspective. All this should also be considered in the drawing up and updating of clinical guidelines and protocols (including in a standardized and explicit manner related unnecessary practices) as well as in the process of their implementation and evaluation. Moving forward in all these lines of work would undoubtedly contribute to decisive improvements in clinical practice, patient safety, and the efficiency of the health system. 

Further studies are needed to better understand this phenomenon in Spain, including the patient’s perception and knowledge, as well as evidence-based clinical practices’ adherence, to estimate the magnitude and impact of low clinical practices in the public health system.

## 5. Conclusions

Patient pressures and defensive medicine are the main determinants for the indication of unnecessary clinical procedures in the surgical field, although the media could also play an important role. Educational campaigns to reduce the number of unnecessary patient requests could have a major impact on this problem, according to the surgeons and anesthetists of our environment.

In addition to identifying unnecessary practices and improving ICC dissemination, it is necessary to identify the frequency of overuse, and estimate the knowledge that surgical professionals have about these practices, as well as the strategies needed to avoid them.

Appendix A: Review—Survey Instrument.

## Figures and Tables

**Figure 1 ijerph-17-03556-f001:**
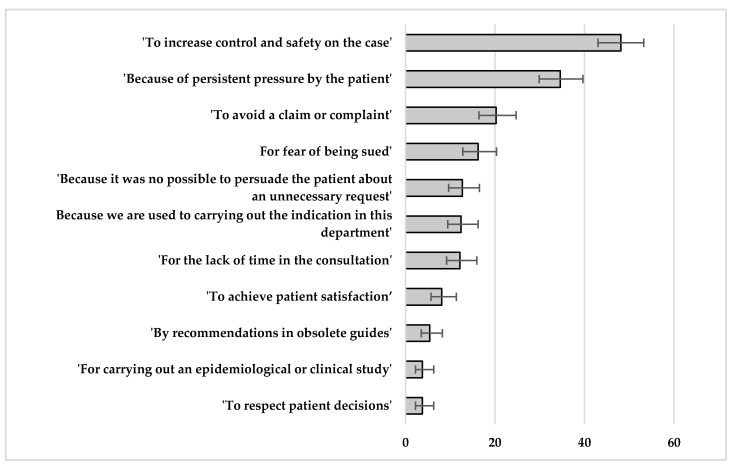
Main reasons why the professionals indicated unnecessary clinical procedures. Percentage of professionals who indicated each reason (including the 95% confidence intervals for each estimation).

**Figure 2 ijerph-17-03556-f002:**
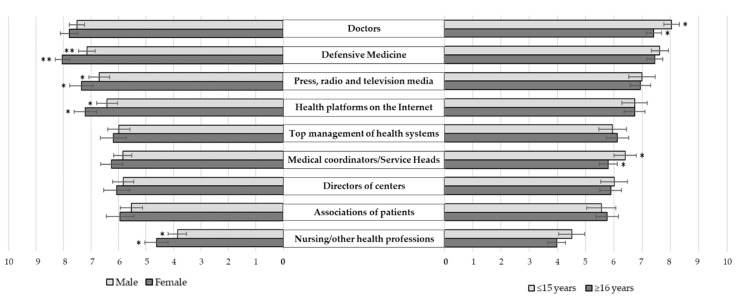
Degree of responsibility regarding overuse referred, according to the professional’s gender and work experience. Scale from 0 to 10, where 0 means “No responsibility” and 10 “Maximum responsibility” (with the expected 95% confidence interval of such percentages). * *p* < 0.050; ** *p* < 0.001.

**Figure 3 ijerph-17-03556-f003:**
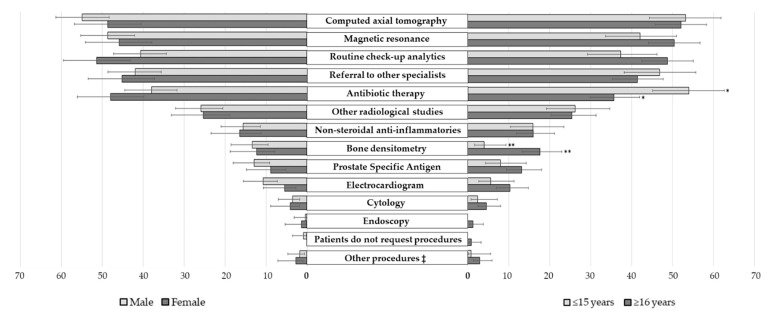
Frequency of patients’ unnecessary procedures requested to surgical doctors, according to the gender and experience of professionals. Percentage of professionals who indicated each reason (with the 95% confidence interval of such percentages). Each respondent could mark as many procedures deemed necessary. ‡: “Surgical treatment”, “digestive tumor markers”, “genetic studies”, “cesarean sections”, “lactose intolerance test”, “electrotherapy and thermotherapy”, “allergy tests”, and” electromyography”. * *p* < 0.050; ** *p* < 0.001.

**Figure 4 ijerph-17-03556-f004:**
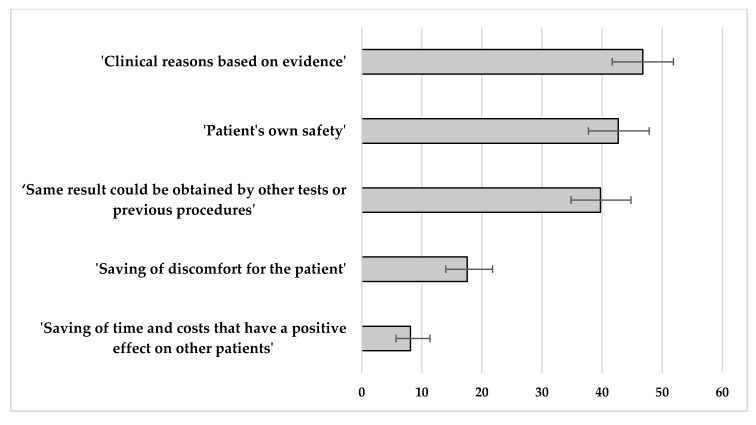
Arguments considered “highly effective” or “very highly effective” to persuade the patient about the inadequacy of medical procedures. Percentage of professionals who indicated each reason (with the expected 95% confidence interval of such percentages).

**Table 1 ijerph-17-03556-t001:** Frequency of request and indication of unnecessary procedures by patients and professionals.

Situation	Frequency	Global		Work Experience of the Health Care Professional Respondent			
				≤15 years		≥16 years	
		*n*	% (95% CI)	*n*	% (95% CI)	*n*	% (95% CI)
Patients request unnecessary medical tests or procedures to doctor	Never	20	5.4 (3.5 to 8.3)	3	2.4 (0.8 to 7.2)	17	7.0 (4.3 to 11.0)
	Monthly	165	44.6 (39.6 to 49.7)	58	46.0 (37.4 to 54.9)	107	43.9 (37.7 to 50.2)
	Almost every week − every day	185	50.0 (44.9 to 55.1)	65	51.6 (42.8 to 60.3)	120	49.2 (42.9 to 55.5)
Doctor indicates unnecessary medical tests or procedures due to pressure from a patient	Never	125	33.8 (29.1 to 38.8)	34	27.0 (19.9 to 35.5)	91	37.3 (31.4 to 43.6)
	Monthly	189	51.1 (46.0 to 56.2)	72	57.1 (48.3 to 65.6)	117	48.0 (41.7 to 54.3)
	Almost every week − every day	56	15.1 (11.8 to 19.2)	20	15.9 (10.4 to 23.5)	36	14.8 (10.8 to 19.8)
Doctor convinces the patient that the unnecessary medical procedures he requests are not appropriate	Never	30	8.1 (5.7 to 11.4)	7	5.6 (2.6 to 11.3)	23	9.4 (6.3 to 13.8)
	Monthly	128	34.6 (29.9 to 39.6)	40	31.8 (24.1 to 40.5)	88	36.1 (30.2 to 42.3)
	Almost every week − every day	212	57.3 (52.2 to 62.3)	79	62.7 (53.8 to 70.8)	133	54.5 (48.2 to 60.7)
Doctor receive patient’s negative or aggressive reaction after an explained refuse of their request	Never	122	33.0 (28.4 to 38.0)	27	21.4 (15.1 to 29.6) *	95	38.9 (33.0 to 45.2) *
	Monthly	149	40.3 (35.4 to 45.4)	63	50.0 (41.2 to 58.8)	86	35.3 (29.5 to 41.5)
	Almost every week − every day	99	26.8 (22.5 to 31.5)	36	28.6 (21.3 to 37.2)	63	25.8 (20.7 to 31.7)

*n*: sample; % (IC95%): percentage (Expected interval of such percentage with a confidence of 95%). * *p* < 0.050. Each respondent could only mark one reply option.

**Table 2 ijerph-17-03556-t002:** Knowledge of some surgical related “Do not do” recommendations among surgical doctors.

Knowledge Measured	Total Correct Answers		Correct Answers, According to work Experience of the Health Care Professional			
			≤15 years		≥16 years	
	*n*	% (95% CI)	*n*	% (95% CI)	*n*	% (95% CI)
Patient’s autoadministration of informed consent before invasive procedure (I)	155	41.9 (36.9 to 47.0)	58	46.0 (37.4 to 54.9)	97	39.8 (33.8 to 46.1)
PSA indication for prostate cancer screening in all men over 55 years old (I)	187	50.5 (45.4 to 55.6)	73	57.9 (49.0 to 66.4) *	114	46.7 (40.5 to 53.0) *
Indication and duration of peri-operative antibiotic prophylaxis for all surgical patients (I)	262	70.8 (66.0 to 75.2)	107	84.9 (77.5 to 90.2) **	155	63.5 (57.3 to 69.4) **
Indication of hair removal only if compromise surgical field visibility (C)	269	72.7 (67.9 to 77.0)	87	69.1 (60.3 to 76.6)	182	74.6 (68.7 to 79.7)
Indication of radiological and blood test for every anesthetic evaluation (I)	308	83.2 (79.1 to 86.7)	111	88.1 (81.1 to 92.7)	197	80.7 (75.3 to 85.2)
Indication of PPI in all hospitalized patients for gastroduodenal ulcus prophylaxis (I)	317	85.7 (81.7 to 88.9)	111	88.1 (81.1 to 92.7)	206	84.4 (79.3 to 88.5)
Indication of weekly infection screening in patients with urinary catheter (I)	343	92.7 (89.6 to 95.0)	118	93.7 (87.7 to 96.8)	225	92.2 (88.1 to 95.0)

PSA: Prostate-Specific Antigen; PPI: Proton Pump Inhibitors; (I): incorrect procedure; (C): correct procedure. *n*: sample; % (95% CI): percentage (Expected interval of such percentage with 95% confidence). * *p* < 0.050; ** *p* < 0.001.

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
