# Peer review of "Low-Value Clinical Practices: Knowledge and Beliefs of Spanish Surgeons and Anesthetists"

_ijerph, 2020, doi:10.3390/ijerph17103556_

Round 1

Reviewer 1 Report

It's an interesting topic presented by authors. But the results show that there is not much significant difference regarding four results (from page 3, line 128:  from "3. Results" to page 7 before line 210, 4. Discussion). There are not clearly present all results. May consider trying using the chart to display the comparisons. 

Since all results are not clearly showing, the discussion is out of focusing.

It's much appreciated if the authors redo the analysis for all results and return to me the new presentation for review again.

Reviewer 2 Report

The article entitles “ Low-Value Clinical Practices: Knowledge and Beliefs of Spanish Surgeons and Anesthetists” clearly discussed about the low value surgical practices and defensive medication in Spain.

As mentioned in the manuscript Patient’s pressure is also one of the reasons for low value clinical practices. Awareness through educational campaigns is needed to avoid unnecessary patient request on tests.

This study was well designed and executed. Results are clearly presented.

However, the authors should have been chosen various countries in addition to Spain.

Any reason for choosing only Surgeons and Anesthetists?

Author Response

Thank you so much for your comments; we appreciate your interest in our work and your suggestions, which help us to improve our manuscript.

Please, find below our answers for each of your observations:

Point 1. The authors should have been chosen various countries in addition to Spain.

Response 1. We are agree, it would be very interesting to extend this research for other countries. The questionnaire was aimed at Spanish professionals because the study has been developed in the context of “No Hacer” (“Do-not-do”, the Spanish “Choosing Wisely” Initiative) launched by the Spanish Ministry of Health, Social Services and Equality. This initiative just collects the most relevant procedures to avoid in health care but, in our country, there is scarce information about their frequencies, so we tried to fill this gap with our research.

Similar questionnaires have been used in other countries, like USA [1] or Italia [2], so we considered interesting to apply also among the Spanish professionals. The questionnaire was sent to medical doctors of Spanish hospitals, in the regions of Madrid, Valencia and Murcia.

  1. Perry Undem Research/Communication Unnecessary tests and procedures in the health care system what physicians say about the problem, the causes, and the solutions results from a national survey of physicians 2014.
  2. Vernero, S.; Giustetto, G. Esami diagnostici, trattamenti e procedure non necessari: risultati e considerazioni da un’indagine sui medici italiani. Recenti Prog Med 2017, 108, 324–332.

To improve this issue, as well others related, we have included this paragraph in the Discussion section (page 9, lines 312 to 316):

On the other hand, our results are limited to a sample of medical professionals in the surgical field; It would be of great interest to be able to investigate all these aspects in other healthcare areas and health professionals in order to fully understand the phenomenon of overuse. In addition, it is necessary to consider possible cultural differences that influence the determinants of overuse, with respect to the results reported by other studies.

Point 2. Any reason for choosing only Surgeons and Anesthetists?

Response 2. The research group SOBRINA is studying low value clinical practices in two different settings: the first is primary care health professionals, with several papers published yet (all of them available at the website www.nohacer.es). The second is focused in the surgical area, including this research (exploring frequency, features and determinants of overuse), and also other two related studies: adequacy of X-ray tests among surgical patients; and claims in surgical patients (frequency, characteristics and economic impact). We tried to cover the overuse in this setting, choosing only Surgeons and Anesthetists, but indeed we agree that expand this study to other medical specialties should be relevant and necessary too.

To clarify this point, we have included this paragraph in the Material & Methods section (page 2, lines 87 to 91):

This study has been developed in the context of the “No Hacer” initiative (“Do-not-do”, the Spanish “Choosing Wisely” Initiative) launched by the Spanish Ministry of Health, Social Services and Equality. In addition, it is part of a project to characterize the phenomenon of overuse in the surgical field, considering its frequency, characteristics and determinants; as well as the economic impact on potentially related claims.

Also, as commented above, we included this paragraph in the Discussion section (page 9, lines 312 to 316):

On the other hand, our results are limited to a sample of medical professionals in the surgical field; It would be of great interest to be able to investigate all these aspects in other healthcare areas and health professionals in order to fully understand the phenomenon of overuse. In addition, it is necessary to consider possible cultural differences that influence the determinants of overuse, with respect to the results reported by other studies.

We enclosed a marked copy of the manuscript, including all of these changes. Please, let us know if you find them suitable, or if you consider other changes to enhance the paper.

Round 2

Reviewer 1 Report

There is some revised presented in the new version for about the result part.

As I said in my first reviewing, it will be much better if the comparison result can be represented by bar chart etc.
